# Exploring midwives' understanding of respectful maternal care in Kumasi, Ghana: Qualitative inquiry

Veronica Millicent D-zomeku[1]*, Bemah Adwoa Boamah Mensah[1], Emmanuel Kweku Nakua[2], Pascal Agbadi[1], Jody R. Lori[3], Peter Donkor[4]

**1** Department of Nursing, College of Health Sciences, Kwame Nkrumah University of Science and Technology, Kumasi, Ghana, **2** Department of Epidemiology and Biostatistics, School of Public Health, Kwame Nkrumah University of Science and Technology, Kumasi, Ghana, **3** School of Nursing, University of Michigan, Ann Arbor, Michigan, United States of America, **4** Department of Surgery, School of Medical Sciences, Kwame Nkrumah University of Science and Technology, Kumasi, Ghana

* vydzomeku@gmail.com

**Data Availability Statement:** All relevant data are within the paper.

**Funding:** This project was financially supported by a grant awarded by National Institutes of Health

## Abstract

### Background

Various aspects of disrespect and abusive maternity care have received scholarly attention because of frequent reports of the phenomenon in most healthcare facilities globally, especially in low- and middle-income countries. Experiences of disrespect and abuse during childbirth may dissuade women from returning for facility-based postpartum services, for antenatal care, and delivery for future pregnancies and births. Midwives' knowledge of respectful maternity care is critical in designing any interventive measures to address the menace of disrespect and abuse in maternity care. However, the perspectives of skilled providers on respectful maternal care have not been extensively studied. Therefore, the present study sought to explore the views of midwives on respectful maternity care at a teaching hospital in Kumasi, Ghana.

### Methods

We employed an exploratory descriptive qualitative research design using an interpretative approach. Data were generated through individual in-depth interviews of fifteen midwives, which were audio-recorded and transcribed verbatim. Open Code 4.03 was used to manage and analyse the data.

### Findings

The midwives demonstrated some degree of awareness of respectful maternity care that comprised of the following components: non-abusive care, consented care, confidential care, non-violation of childbearing women's basic human rights, and non-discriminatory care. However, midwives' support for disrespectful and abusive practices such as hitting, pinching, and implicitly blaming childbearing women for mistreatment suggests that midwives awareness of respectful maternity care is disconnected from its practice.

through the Fogarty International Center under Award Number K43TW011022. The content is solely the responsibility of the authors and does not necessarily represent the official views of the National Institutes of Health. The funders had no role in study design, data collection and analysis, decision to publish, or preparation of the manuscript.

**Competing interests:** The authors have declared that no competing interests exist.

## Conclusion

In view of these findings, we recommend frequent in-service training for midwives and the institutionalization of regular supervision of intrapartum care services in the healthcare facility.

## Background

The rise in facility-based deliveries with skilled providers in low-and-middle-income countries (LMICs) has resulted in decreased maternal and neonatal morbidities and mortalities [1–5]. In recent years, frequent accounts of facility-based disrespect and abusive care (D&AC) are undermining the encouragement of childbearing women to access intrapartum care services in healthcare facilities [6–10]. D&AC violates childbearing women's rights to quality maternity care, including dignified care and freedom from discrimination [11]. Preventing and eliminating facility-based D&AC in LMICs requires that countries implement a scalable, sustainable, and cost-effective solution. The World Health Organization (WHO) statement on addressing D&AC suggests that ensuring and integrating respectful maternity care (RMC) in obstetric care—pregnancy, childbirth, through postnatal care—is the appropriate solution to pursue [11]. RMC refers to "care organized for and provided to all women in a manner that maintains their dignity, privacy and confidentiality, ensures freedom from harm and mistreatment, and enables informed choice and continuous support during labour and childbirth" [12]. RMC has been measured through systematic labour observations, cross-sectional surveys using standardized questionnaires as well as through in-depth interviews with maternity care providers and postpartum women [8, 13–16]. In these studies [8, 13–16], the RMC metrics focus on childbirth experiences in the following domains: physical, verbal, and psychological abuse, discriminatory care, vaginal examinations, companionship, pain relief, choice of alternative birthing position, and health system drivers.

Studies have shown the effectiveness of RMC interventions in reducing D&AC and maternal and neonatal deaths in LMICs [17, 18]. Interventions providing mentorship and training that transformed negative provider attitudes and improved the interpersonal communication skills of caregivers, along with interventions equipping facilities with efficient monitoring systems, and interventions educating childbearing women on their rights have had the most positive effect [18]. One intervention integrated specific components of RMC, like dignity, respect, communication, autonomy, and supportive care, into a simulation training [17]. The simulation training was designed to improve the identification and management of obstetric and neonatal emergencies [17]. It was piloted in the East Mamprusi District in the northern region of Ghana [17]. The findings showed that RMC training workshops have the potential to improve childbearing women's childbirth experiences in LMICs [17].

To prevent and eliminate facility-based D&AC during childbirth, WHO recommended five key actions, one of which is to generate data related to respectful and disrespectful care practices [11]. In line with this, several studies have generated knowledge on the perspectives of childbearing women and skilled providers on D&AC [6, 19–23]. The few studies that explored the views of skilled providers on RMC were from countries other than Ghana [24, 25]. Thus, the present study seeks to explore midwives' awareness of RMC within the context of labour and childbirth in a tertiary health facility in Kumasi, Ghana. The aim of the study originates from the dearth of literature coupled with the frequent reports of D&AC by mothers in many healthcare facilities across the country [6, 26–28]. Documenting the thoughts of midwives on

RMC will help the national midwifery and nursing council and training schools to address existing gaps in knowledge and practice of RMC by restructuring the national midwifery curriculum. It will also provide further qualitative insights into disrespectful and abusive practices in maternity care.

## Methods

### Research design

A descriptive qualitative design using an interpretative approach was employed to explore midwives' awareness of respectful maternity care. This approach was used because it permits the authors to explore and document midwives' description and interpretation of actions that they consider to be respectful maternity care [29]. This study is part of a larger qualitative research study exploring the feasibility of changing the culture of disrespect and abuse in maternity care.

### Study setting

The study setting was a tertiary health facility in Kumasi, located in the Ashanti Region, the central part of Ghana. The study setting was chosen because of the major role it serves in Ghana's healthcare delivery. The tertiary health facility in focus serves patients across the country and has a bed capacity of approximately 1,200 and a staff strength of about 3,000. It is the main referral hospital serving patients with a diverse ethnic and socioeconomic background mainly from six political and administrative regions in Ghana—the Ashanti, Brong Ahafo, Western, the three Northern regions (Northern, Upper East, And Upper West)—, and neighbouring countries. It has twelve (12) clinical directorates, one of which is the maternal and child health directorate (MCH-D). The MCH-D has two main units—the general ward and the special ward. The general ward is an open ward the use of which is covered under national health insurance, and the special ward is a private ward which is accessed at an extra cost.

### Population, sampling, and sample size

The study population was composed of midwives in this tertiary health facility in Kumasi, Ghana. The inclusion criteria for the study were midwives who have had at least one year of professional practice and were working on the labour ward. Purposive sampling ensured that participants who met the inclusion criteria and gave written consent were enrolled. The second author and two research assistants (RAs) recruited the study participants.

### Data collection

A semi-structured interview guide and face-to-face in-depth interviews were used to generate the data. The guide was developed based on a respectful maternal care module (RMC-M) developed by the first author in her preliminary studies [30]. The researcher and the RAs asked probing questions to acquire an exhaustive and in-depth understanding of the participants' awareness of RMC. Data collection started on 3rd January 2019 and ended on 25th February 2019. The interview guide was pre-tested with three (3) midwives working at the maternal unit of the Kwame Nkrumah University of Science and Technology Hospital (which is different from the study setting), Kumasi, to ensure the appropriateness of the guiding questions. The interviews were conducted in English, but participants were permitted to express themselves in Twi, the native language of the region. The authors approached the midwives individually, discussed the study and obtained written consent prior to the interviews. Four

interviews were conducted every week by the second author, assisted by the two RAs, to allow for transcription and coding to ascertain patterns in the responses. The interviews ended with the 15[th] participant, as no new information emerged [31, 32]. The interviews were conducted by the second author (ABBM), a qualitative researcher with clinical and academic experience in women's health and maternal care. As a researcher, ABBM speaks and writes both Twi and English languages. The interviewer does not work at KATH; hence she had no direct influence on the study setting and the participants. The interviews language was English. The interview duration was 50–80 minutes, and the interviews were audio-recorded with the participants' consent. Date, time and venue (enclosed office at the study setting) of the interviews were scheduled to be suitable for participants. During the interview sessions, the two RAs took field notes of non-verbal cues and other relevant observations.

## Data management and analyses

The first and second authors and the RAs thoroughly listened to the audio files for accurate verbatim transcription. The first author (VMD) has same educational and research experience as the second author in addition to midwifery education. The third and fourth authors (with broad academic and research backgrounds in public health and biostatistics) proofread the transcribed interviews to ensure that participants' views were precisely captured. Anonymity was ensured by serializing each transcript file, and the transcripts were kept in a secured folder on the laptop of the principal investigator. Open Code 4.03 was used to manage and analyze the data presented under predetermined subthemes. The D&AC project was created in the software, and the transcripts were saved as text files and imported into the project folder. The first and second author analysed the data independently and this was independently confirmed by the fourth author and validated by the third author. The codes generated during the analyses were synthesized into predetermined subthemes.

Confirmability, replicability, dependability, and authenticity were the trustworthiness criteria followed in the study [33]. Confirmability was achieved through member checking to help ensure participants' realities were accurately presented. Member checking was done by allowing four of the participants review the data to ensure accurate presentations of their views before the final conclusions were drawn [33]. Also, independent analysis by the third and fourth authors mentioned earlier contributed to analytical rigour and confirmed the findings. To ensure replicability, the authors provided a detailed description of the research design and data collection methods and setting as well as the background of the participants and the potential replication of the study by future researchers. Through peer debriefing and strict adherence to the study protocol, the trustworthiness of the data was further ensured.

## Ethics approval and consent to participate

The Committee on Human Research, Publication, and Ethics (CHRPE) at the Kwame Nkrumah University of Science and Technology (KNUST) gave the ethical clearance for the study (CHRPE/AP/181/18), and the managerial board of the teaching hospital gave the institutional approval for the study (RD/CR17/289). Participants were briefed on the study and their rights to voluntary participation and withdrawal from the study with no consequences. Only participants who consented were involved in the study. The participants gave their consent to the interview, the audio recording of the session, and the publication of findings. Participants' confidentiality was ensured by conducting the interview in an enclosed office. Information that could reveal the identities of the participants were excluded from the transcripts to ensure participants' anonymity.

## Findings

### Demographic features of participants

The midwives were, on average, 33 years old, with a range of 31–48 years. They had engaged in professional practice for an average of eight years (range = 6–11 years). Seven participants obtained a bachelor's degree in midwifery, while nine participants received a diploma [diploma is a post-secondary qualification that's lower than a bachelor's degree in Ghana's educational system]. Only one of the midwives was a Muslim, and the others were Christians. Eleven were currently married. Those with children (n = 10) had an average of 2.3 living children (range = 1–3).

### Respectful and non-abusive maternity care (RMC)

The findings are presented under three main themes: awareness of respectful maternity care (RMC), motivations for RMC, midwives' recommendation for optimal RMC, and disconnect between awareness and practice of RMC.

### Awareness of RMC

All the midwives in the current study demonstrated some awareness of respectful and non-abusive maternal care. The sub-themes associated with the description of RMC include respectful care, non-abusive care, ensuring childbearing women-centred care, and respecting childbearing women's rights.

**Respectful care.** In describing respectful care, the midwives mentioned certain nursing care measures that the childbearing women may require to cope with the process of labour. Some of these measures include consent seeking, clinical procedure explanation, and providing sacral massage.

*You [the midwife] are there to support that childbearing woman at that moment, so, you assist in sacral massage, and you give her encouraging words in order that the childbearing woman will not feel that she is in that alone. . .there are a lot of childbearing women in there, so we are in as their support to go through their labour and the pain. . .So, we are the caregivers and we are also the supporters . . .we are giving them the support during the labour.* [Midwife 001].*

*You must show love and welcome her, so she feels she has come home. In that way, even if she came with some anxiety, she will become relaxed. . .I consider it [consent seeking] as respect because if you are just there, and out of nowhere, someone tells you to turn to get an injection, even if you are not afraid, you could be scared a little. But if a worker comes and explains a procedure to you, for instance, because of the headache you are experiencing, this injection will take absolute care of it, and then she seeks your consent to give you the injection, it shows respect. . . .* [Midwife 004].*

**Non-abusive care.** The midwives described and gave examples of non-abusive care as care that does not compromise the physical, psychological dignity, and wellbeing of childbearing women. The key phrases in their descriptions include reassuring childbearing women, ensuring childbearing women's comfort, avoiding the use of harsh or abusive language, and the use of non-verbal communication.

*Non-abusive [care] is rendering care that will not make the childbearing woman feel bad. Yeah, care that will make the childbearing woman feel comfortable at your ward. Yeah. You*

*ensure by the way you talk to the childbearing woman nicely and encouraging, reassuring, or make the childbearing woman comfortable.* [Midwife 007].

*Non-abusive care is the care you give to childbearing women without using abusive words, like insults or shouting or using non-verbal languages to communicate with childbearing women.* [Midwife 013].

**Client-centred care.**   In describing RMC, most of the midwives mentioned and offered practical examples of an individual- or client-centred care. It involves taking into consideration the childbearing women's unique experiences, feelings, and response to labour and care.

*Well, there are a lot of clients who come here. And each one has peculiar characteristics. Let's use labour as an example. Someone will go into labour quietly. If you don't get close to her, by the time you realize, she will deliver the baby without you. But someone else will shout and wail from 2cm till the end. So, each person is different. . .you should be able to care for each childbearing woman devoid of discrimination. You first determine the peculiarities, and then give her appropriate care that will help make the childbearing woman better.* [Midwife 004].

**Respecting childbearing women's rights.**   The researchers explored the awareness of midwives on the rights of childbearing women during care. They indicated that childbearing women, first and foremost, have the right to RMC. Other rights to which midwives believed childbearing women were entitled include the right to confidentiality and anonymity of health status, the right to consented care, the right to have access to relatives during the stay at the hospital, the right to health information, the right to privacy, and the right to choose from treatment options.

*The right to care, the right to information, the right to privacy, the right to the best of care and the right to consent to any procedure supposed to be performed on her, and the right to get an explanation concerning any procedure supposed to be performed on her . . .Ideally, before you administer any drug to a childbearing woman, the first thing you do is to explain what the drug is going to do and then the consequences attached to it.* [Midwife 002].

*. . .A single room like here [pointing to a room], and she wants a relative to visit or be around her, suppose she's never slept alone in a room before, she has the right to have a relative be around as long as there is room for that to happen. . .Also, there are certain kinds of information, though she may have come with her husband, she has a right to demand that certain information remains between you and her; her husband doesn't even have to know. She also has that right. . .And then, for instance, a childbearing woman can ask for an alternate, say an oral medication instead of an injection because she may not like injections. For that too, she has a right because she can't take an injection and as long as an alternative exists that she can orally ingest, she has that right. Also, any information concerning her care, she has every right to have access to it. . .* [Midwife 004].

## Motivations for RMC

The midwives had certain motivations for providing childbearing women with RMC. Their motivations centred on the perceived connection between the mother's treatment and neonatal outcomes, the midwifery profession as God's calling, the economic, social, and psychological benefits associated with RMC, and the experience of labour.

**Perceived connection between the mother's treatment and neonatal outcomes.** The midwives interviewed had experiences that compromised or enhanced the wellbeing of child-bearing women. Thus, they understood childbearing women's reactions to both abusive and non-abusive care. From the findings, midwives were motivated to provide RMC because of the perceived connection between how the mother is treated and neonatal outcomes. According to the midwives, childbearing women remain appreciative to them due to the RMC they received.

*The most important thing is to deliver a healthy baby, but that ultimately depends on the kind of care rendered to the mother. So, if the childbearing woman is wet, she must be wiped and changed. . . .The baby inside the mother also experiences whatever you do for the mother. The baby needs to enter this world with joy and that depends on the conditions the mother experiences. So, if you educate her to do everything correctly, the pushing becomes exceptional when the time is due; the mother has the energy to push and the baby is strong.* [Midwife 004].

**God's calling.** Some of the midwives noted that God called them to serve as midwives, and that motivates them to provide optimal RMC. They had the following to say:

*I believe God has given me this opportunity [serving as a midwife] so I won't let Him regret giving me the blessing.* [Midwife 004].

*What I want to share with them [colleague midwives] is that this job is a calling, it's from God and if you do it well, the blessing is even here on earth. And some people say when you do well here, when you get to heaven you will get it back, but we get the reward, whether good or bad, we get them here.* [Midwife 005].

**Economic, social, and psychological benefits.** The midwives indicated that the benefits they have reaped due to providing RMC encouraged them to continually provide RMC. The benefits were economic, social, and psychological in nature. Paramount among the benefits they accrued from providing RMC was emotional appraisal from postnatal women and their relatives.

*They [the childbearing women] really remember whatever you do to them. Sometimes you meet them in town, and they call you and you feel happy. They sometimes say, "This is your grandchild," and you are like, wow! "I haven't even given birth yet, but I have a grandchild." It's beautiful when they call you. Sometimes, call you to find out how you are doing. . . [Sometimes, they will say] "Oh bring your clothes and let me sew them for you."* [Midwife 001].

*Sometimes when the client comes, in my experience when they come, and you are free with them, you talk to them very nicely, even when they have given birth, they are coming for review, they see you, they just approach very nicely and they 'Oh, God bless you. This midwife, when I came to deliver, she was exceptionally good to me. This midwife has a welcoming demeanour'. It is very good when people use such words to describe you. Even the relatives, when they come, they talk to us and we receive them properly.* [Midwife 009].

**Midwives' experience of labour.** One midwife, who had an experience of giving birth, noted that experiencing labour puts you in a position to provide RMC.

*I once heard that if you'd never given birth before, you were never allowed into midwifery. But there came a time when they needed more people and all these young people were allowed in. Because if you've ever had babies [you will appreciate the need to provide good care to child-bearing women going through similar labour experience] . . .when I was in the labour ward, my colleagues always complained that I had too much tolerance for bad behaviour.* [Midwife 004].

## Recommendation for optimal RMC

**Labour and pain management education.** Some of our participants suggested that, to ensure optimal RMC, during antenatal care visits, childbearing women should be thoroughly equipped with information on the phases of labour and pain management. They believe that this education will help childbearing women to cope with and manage their pains during labour and make them cooperative with caregivers when admitted at the facility.

*. . .when they are pregnant, we should give them every education they need to know, every information they need to know. Sometimes, they don't even know how to even attach the baby to the breast, which is very bad. But if from antenatal she is knowing that this is how labour is, this is how when the baby is about coming, this is how it is, this is how you have to attach your baby to the breast, "even we ourselves, we won't get tired/frustrated" because she knows so she will act according to that information she is having. . .. [Midwife 001].*

*Sometimes, they need to be talked to, especially during the antenatal. We need to educate them on what they are supposed to do when they are in labour. If we do that very well, when they come to labour, they will know what they are supposed to do. . .the person [childbearing woman] comes to labour not knowing anything about the first stage, second stage or third stage. All she knows it; she has come to give birth. We have to educate her, so she understands what she is coming to accomplish. So next time she is coming, she knows this sister said when I do get here, I will do this, I will do this. So, don't say 'she came to deliver, and it is done', no. sometimes you have to educate her on the baby.* [Midwife 008].

**A disconnect between awareness and practice of RMC.** The views of midwives were sought on situations where force should be applied during intrapartum care. This question was informed by findings from previous work by the first, third, and fifth authors involving midwifery students who justified the application of force or hitting childbearing women during labour [34]. The midwives who responded to the questions mentioned that hitting the thighs of the childbearing women in the second stage of labour helps them push and deliver babies safely. Their views on hitting childbearing women during labour carried moral overtones, as they judged the action to either be wrong or right.

*when the head is crowned, then you have to hit the childbearing woman [in labour]. If you don't, she will make you lose the baby. . .and people will blame you or call you names. Before that, you have to advise the childbearing woman to push. But there are some that you will advise, but they will not take it. But once you hold the thigh like this [demonstrating pinching], they become conscious. And when they are done, they even apologize to you. . .at that particular time, if any midwife does this, she hasn't done anything wrong, because at that time, either we save the mother, or we save the baby.* [Midwife 008]

Some midwives were not sure whether hitting childbearing women has any scientific or clinical merit, but they felt such a practice was not right.

*. . .Well, it's bad to hit childbearing women. . .I. . .only [do it] when conducting delivery and she is closing the gap and I hit in-between the thighs 'open up!', aha, that's the only time I hit a childbearing woman, and it is not hitting deliberately. . .*[Midwife 003]

*. . .I've not done some before but I think it's not good to hit a childbearing woman.* [Midwife 008]

## Discussion

Midwives in this study have some awareness of respectful maternal care. Their support for disrespectful and abusive practices such as hitting, pinching, and implicitly blaming childbearing women for mistreatment suggests that midwives awareness of RMC is disconnected from the practice of RMC.

Other studies in healthcare facilities in Ghana have revealed that childbearing women are usually mistreated, and the midwives in our study and in others justified mistreatment by arguing that childbearing women should not be overly pampered as this may lead to non-compliance and death of their babies [6, 26, 28]. The repetitiveness of this finding in our study involving practising midwives and that of previous work by the lead author [34] involving student midwives suggest that this belief of engaging in mistreatment with a life-saving goal in mind is pervasive and deserve to be immediately addressed. Given the goal of this sub-study to inform intervention, we emphasised the disrespectful and abusive nature of such a belief and its dangers on the dignity and welfare of childbearing women in our training manual and during the intervention.

The midwives made reference to intrinsic and extrinsic motivations for providing RMC. These motivations include an emotional appraisal, gifts from postpartum women, and the regarding of midwifery profession as God's calling. This finding is congruent with findings obtained from studies in Ghana, Burkina Faso, and Tanzania [35, 36]. These studies indicated that midwives were motivated to provide RMC because of appreciations received from the community, positive feedback from superiors, perceived government and NGO support, and their sense of duty towards the delivery of healthy babies and God [35, 36]. Given that some of the midwives in our study were motivated by emotional appraisals, the facility can institute an award that recognizes the provision of good maternity care practices through clients' reviews and managerial supervision systems. Additionally, punitive measures for mistreating childbearing women should complement the RMC recognition award while pursuing the search and elimination of other potential factors that may impede the practice of RMC in the facility.

Midwives also mentioned that educating childbearing women about labour and pain management will ensure optimal RMC during childbirth. This recommendation suggests that midwives rationalize disrespectful care as being a consequence of women's poor attitude and poor pain management. It is laudable to provide quality pain management education to childbearing women during antenatal care visits; however, the training of midwives to learn to provide childbearing women with RMC practices regardless of women's knowledge of labour expectations is an essential strategy for promoting respectful maternity care [17]. The recommendation by the midwives clearly suggests that they finds it difficult to practice RMC in situations where childbearing women are unable to bear with the labour pain and the midwives think that the attitude of the women will lead to the death of the baby.

Research on the implications of the disconnect between knowledge and practice in healthcare are amply documented, with complex systems such as motivational, institutional, and sociocultural factors cited as barriers to applying knowledge [37, 38]. Given that disrespectful and abusive care within the context of labour and childbirth has recently started receiving critical scholarly appraisal, not many studies have documented the disconnect that exist between

midwives' knowledge of respectful maternity care practices and what they practice in the labour ward. Our study highlight this disconnect. Therefore, this finding call for research opportunities that probe this issue further and explore provider and institutional level factors that may be driving the disconnect in knowing what to do to promote optimal labour and childbirth experience to childbearing women and the practice of disrespectful care. While disrespectful and abusive practices in maternity care is never acceptable under any circumstance, the call for further research to explore drivers can as a matter of urgency examine whether health system challenges contribute to healthcare workers' inability to practice RMC in the study setting. With findings from studies guided by these recommended research questions, policymakers will be equipped to design programs and policies that holistically address disrespectful care in intrapartum care.

## Limitation and strength

One limitation of this study is the issue of generalizability of findings for other midwives in the teaching hospital itself, the region of the study setting, and the entire country due to the small sample of midwives interviewed. Another limitation is that we were unable to confirm whether the reported best practices are actually being implemented by midwives in the facility. Future studies could build on this work by assessing the quality of care delivered in this setting through clinical observation. An important strength of this study is the richness of the data; frontline maternity care providers—midwives—gave their opinions on how they think and interpret RMC care, which is critical in both understanding and improving this care.

## Conclusions

The objective of the study was to explore midwives' experiences of and views on respectful maternal care. The findings revealed that the midwives demonstrated some awareness of respectful maternal care, and their support for disrespectful and abusive practices such as hitting, pinching, and implicitly blaming childbearing women for mistreatment suggests a disconnect between awareness and practice of RMC. We recommend that there should be frequent in-service training for midwives as well as the institutionalization of regular supervision of intrapartum care services and an award scheme recognizing midwives who engage in quality maternity care in the healthcare facility.

## Supporting information

**S1 File. Interview guide.**
(DOCX)

## Acknowledgments

We acknowledge the midwives who participated and shared their views and experiences on RMC. Pre-Publication Support Service (PREPSS) supported the development of this manuscript by providing pre-publication peer-review and copy editing.

## Author Contributions

**Conceptualization:** Veronica Millicent D-zomeku, Jody R. Lori, Peter Donkor.

**Data curation:** Veronica Millicent D-zomeku, Bemah Adwoa Boamah Mensah, Emmanuel Kweku Nakua, Jody R. Lori, Peter Donkor.

**Formal analysis:** Veronica Millicent D-zomeku, Bemah Adwoa Boamah Mensah, Emmanuel Kweku Nakua, Pascal Agbadi, Jody R. Lori, Peter Donkor.

**Funding acquisition:** Veronica Millicent D-zomeku, Jody R. Lori, Peter Donkor.

**Investigation:** Veronica Millicent D-zomeku, Peter Donkor.

**Methodology:** Veronica Millicent D-zomeku, Peter Donkor.

**Project administration:** Veronica Millicent D-zomeku, Jody R. Lori, Peter Donkor.

**Software:** Pascal Agbadi.

**Supervision:** Veronica Millicent D-zomeku, Jody R. Lori.

**Writing – original draft:** Veronica Millicent D-zomeku, Bemah Adwoa Boamah Mensah, Emmanuel Kweku Nakua, Pascal Agbadi, Jody R. Lori, Peter Donkor.

**Writing – review & editing:** Veronica Millicent D-zomeku, Bemah Adwoa Boamah Mensah, Emmanuel Kweku Nakua, Pascal Agbadi, Jody R. Lori, Peter Donkor.

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
