## [Decision Letter · Decision Letter 0]

29 Aug 2019

PONE-D-19-19693

Exploring midwives’ understanding of respectful and non-abusive maternal care in Kumasi, Ghana: Qualitative Inquiry

PLOS ONE

Dear Mr. Agbadi,

Thank you for submitting your manuscript to PLOS ONE. After careful consideration, we feel that it has merit but does not fully meet PLOS ONE’s publication criteria as it currently stands. Therefore, we invite you to submit a revised version of the manuscript that addresses the points raised during the review process.

Please address each of the specific comments raised by the two reviewers. 

We would appreciate receiving your revised manuscript by Oct 13 2019 11:59PM. To enhance the reproducibility of your results, we recommend that if applicable you deposit your laboratory protocols in protocols.io, where a protocol can be assigned its own identifier (DOI) such that it can be cited independently in the future. For instructions see: http://journals.plos.org/plosone/s/submission-guidelines#loc-laboratory-protocols

We look forward to receiving your revised manuscript.

Kind regards,

Britt McKinnon

Academic Editor

PLOS ONE

Journal Requirements:

a) Did participants provide their written or verbal informed consent to participate in this study?

3. Please include a copy of the interview guide used in the study, in both the original language and English, as Supporting Information, or include a citation if it has been published previously."

4. We noted in your submission details that a portion of your manuscript may have been presented or published elsewhere: "The methods section was taken from a manuscript submitted to the BMC Pregnancy and Childbirth Journal. This does not constitute dual publication because the objectives of the two papers are different."

Reviewers' comments:

Reviewer's Responses to Questions

**Comments to the Author**

1. Is the manuscript technically sound, and do the data support the conclusions?

Reviewer #1: Partly

Reviewer #2: Partly

2. Has the statistical analysis been performed appropriately and rigorously? 

Reviewer #1: N/A

Reviewer #2: N/A

3. Have the authors made all data underlying the findings in their manuscript fully available?

Reviewer #1: Yes

Reviewer #2: No

4. Is the manuscript presented in an intelligible fashion and written in standard English?

Reviewer #1: Yes

Reviewer #2: Yes

5. Review Comments to the Author

Reviewer #1: Thank you for the opportunity to review this paper. The topic of respectful maternal care is interesting, and timely, and as the authors point out – it is important to capture the perspectives of care providers before designing interventions aiming to improve respectful maternal care. Overall, the paper is well written (some minor suggested edits are detailed below), and most statements are well substantiated with appropriate lit review and citations. However, in its current form, I have a few concerns about the methodology, analytical approach, and interpretation of some of the findings. Please refer to my detailed comments below; they are intended to guide further revisions of the manuscript.

Abstract

- While ensuring respectful maternal care is a critical goal in its own right, the rationale for this work may be further strengthened by stating the downstream implications of disrespect/abuse (for example, negative experience during childbirth may dissuade women from returning for facility-based postpartum services or from ANC/delivery for future pregnancies and births). It may be worth adding this rationale to the abstract if space allows.

- Suggest revising the methods to a more active voice (i.e. “we employed a phenomenological study design using in-depth interviews…”)

- What was the general analytical approach? If space allows, a brief sentence describing how transcripts were analyzed would be helpful.

- Line 32: What is meant by “gap in knowledge on evidence-based care”? This reads a little vague and the link to respectful maternal care is unclear. Could the authors be more specific here?

Background:

- Line 39: suggest changing “recent times” to “recent years”

- Line 40: “undermining the purpose of encouraging childbearing…” is a little wordy and could be simplified.

- Lines 41- 42: Suggested edit: “rights to quality maternity care, including dignified care and freedom from discrimination”

- One thing I’m missing from the background/intro section is some indication or comment on why D&AC is so highly prevalent (is it because providers are overburdened, underpaid, etc. or is this completely lacking in medical training, etc. What does the evidence say?)

- Line 50: nice review of relevant literature, but could the authors specify which outcomes are typically used to measure RMC? Line 60, for example: how are “women’s childbirth experiences measured?” Qualitatively? Or are there standard quantitative indicators? Including some detail on the metrics used will be helpful for readers not familiar with this area of work.

- Line 68: “evidence of incidences” is a bit awkward; could this be re-phrased?

- Line 70: suggest editing “further evidence of non-evidence based…” to “further qualitative insights into…”

Design

- I think an overall “Methods” heading would be appropriate before other sections (before “Design”)

- Stating that a phenomenological approach was employed is a bit vague. Suggest to elaborate with a few sentences to explain which phenomenological methods were applied, with appropriate citations.

- What were the specific research questions for this study? The overall aim (understanding midwives’ perspectives of RMC) reads a bit vague.

Setting

- What are the demographics of most maternity care patients in this facility? Are they mostly lower SES, are they diverse in terms of ethnicity, etc.? Some additional detail on the clients/patients served would be helpful.

Population, sampling and sample size

- Purposive sampling: what kind of expertise was considered for interviewee selection? Don’t all midwives have relevant expertise?

- How were the study participants recruited? (approached by researchers directly?)

- Line 93: Sentence needs revising (“All participants who officially…”) the second clause about data saturation is out of place.

- How many providers were invited to participate? And of those, how many agreed to participate?

- The details about data/theme saturation should be moved to Data management and analysis (seems out of place in the sampling section)

Data collection

- How was the interview tool developed? Did researchers draw on any existing theory/frameworks to select relevant questions and probes?

- What settings were “suitable” for participants? Were most conducted within facilities?

Data management and analysis

- Were there any other quality assurance protocols in place? (e.g. de-briefing sessions post-interview, cross-checking transcripts?)

- Line 112: can delete “a qualitative data management software” (implied)

- How were codes developed? Was this an iterative process – e.g. did they emerge from the data or did researchers apply codes based on topics covered in the interview guides?

- Was coding done as a team? How was consensus/agreement on coding reached?

- Line 118: Consider deleting the “trustworthiness/rigor” sub-heading and just include this paragraph under analysis. Alternatively, the sentences in this section could be re-distributed to the relevant sub-headings that precede this section.

- Line 121: What is meant by “member checking”; please define and elaborate and cite a methods source, if appropriate.

- “Independent analysis of the data” is quite vague – what exactly did authors do to cross-check/compare analytical rigor? (particularly when multiple analysis/coders were involved)

- Line 123: The statement about transferability is not convincing; providing sufficient detail is not sufficient to confirm transferability; suggest re-phrasing this.

- Some of the statements about adherence to the protocol and trustworthiness of data may be stated later (discussion section) in a strengths/limitations paragraph

- I think some reflection is needed on researcher positionality relative to interviewees (this is critical information for assessment of qualitative studies).

Findings

- Line 142: what was the range in midwives’ years of practice? Were they all at the same level/rank?

- Line 148: suggest to change to active voice

- It would be helpful to see some examples of questions/probes used to elicit perspectives on RMC (could include the interview guide as a supplementary file, or at least give examples)

- Table 1 - It is not clear to me whether the themes and sub-themes truly “emerged” from the data organically, or if these topics were specifically probed (was the interview guide organized around these themes/sub-themes from the start?); I assume the latter, but this should be stated explicitly if so (and not labelled as “emergent”)

- This reads more as a descriptive, thematic analysis (which is fine) – where does the phenomenological component come in? Phenomenology means different things to different researchers; as currently described, I’m not convinced it’s an appropriate label for the methodology used here.

- Line 157-159: no need to duplicate information presented in table 1 and subsequent sub-headings (sub-themes are already clear in the table and deleting will save word count)

- Line 163: “these are illustrated…” can be deleted (implied that the quotations are meant to illustrate preceding statements)

- I think the second quote is a better illustration of “respectful” care; are comfort measures synonymous with “respect”? Or just good clinical practice? Perhaps including definitions from the literature would be helpful under each of these types of care headings

- A simpler coding system for the quotes could be used (since all participants are midwives, no need to state midwife every time)

- Line 178: “encapsulated providing… “ this sentence is wordy; could be simplified.

- Line 181: “desisting from” – change to “avoiding the use of...”

- Were providers asked specifically to define/explain their understanding of these phrases (e.g. non-abusive care?)

- Line 191: The first quote doesn’t add much (could delete or move to respectful care section), but the second is a nice illustration of women-centered care.

- The text in lines 204-206 could be condensed

- Line 211 (“the following are some of the views”); these lead in sentences are a bit bulky and take up precious word count. Is there a way to integrate relevant quotes into the text more smoothly?

- Lines 228-229: again, it would be helpful to have a sense of which questions were asked

- Line 227 down: the section on hitting women during active labour is important but I’m confused as to why this is included under “evidence-based care” heading and not, for example, respectful or non-abusive care. The link to evidence-based care seems like a stretch – perhaps the authors can better articulate how this component was conceptualized, or consider removing it completely.

- Line 269-280: this content seems somewhat out of place, and doesn’t flow from the preceding paragraphs. The authors may need to clarify how these illustrations specifically relate to RMC.

- Lines 287-207: my interpretation of this quote is that the midwife is motivated by the perceived connection between how the mother is treated and neonatal outcomes (more so than the wellbeing of the woman); perhaps revisit this interpretation or select a different quote?

- Page 16 – the selected quotes are appropriate and interesting, but three may not be necessary; could condense.

- Line 337-338: this text is not needed

- Line 339 down (labour and pain management education): what’s interesting here is that there’s an implicit blaming of women for the abusive/disrespectful treatment (they don’t know how to do what is expected of them during labour, so midwives become tired and frustrated); if only they knew better… Shouldn’t respectful care be the standard, regardless of women’s knowledge of labour expectations? This might be something to consider elaborating on in the discussion/interpretation (you speak to it briefly in lines 385-387); it’s one of the more interesting and novel findings presented.

- It occurs to me, when I get to the end of the Findings section, that there was no clear framework used to guide analysis/interpretation. Are there standard conceptual frameworks and definitions of RMC that could be usefully applied in this paper? I think framing/structuring the results around a current theoretical models/conceptual framework would strengthen the paper.

Discussion (singular, remove “s” from heading)

- First sentence can be deleted (actually, the whole first paragraph could be omitted)

- Lines 360-363: Based on the findings presented, I’m not convinced that they completely understood each component of RMC; perhaps re-phrase this statement to be less definitive.

- There is only one example of “non-evidence based care” given in the results section (hitting), so drawing conclusions about gaps in midwives’ knowledge is not well substantiated.

- Overall, the discussion is quite sparse and would be strengthened with more critical interpretation of the findings, and emphasis on what this study specifically adds to the growing body of work in this area.

- Additional recommendations for policy and practice would also strengthen this section

- To what extent are these ideals (i.e. aspects of RMC identified by midwives) actually practiced? Did the transcripts give any indication of whether respectful care is actually implemented? It is possible that midwives have good knowledge/awareness of what it means to provide respectful care, but this doesn’t mean best practices are actually implemented. This would be important to comment on in the discussion section.

- Line 393: delete “obvious”

- Line 394: Unclear how “exploratory nature” of the study limits its generalizability (this has more to do with the small sample size and is typical of qualitative studies of this nature)

- Perhaps one limitation is that you are unable to confirm whether the reported best practices are actually implemented; future studies could build on this work by assessing the quality of care delivered in this setting (via clinical observation or by interviewing women receiving care)

- Line 395-397: re “bias” this is not so much a limitation (you did not intend to assess objective measures of RMC delivered, but rather the midwives’ lived experience/personal opinions and viewpoints). I think this section could be re-considered more critically, and then revised.

- Line 409: “nation-wide cross-sectional quantitative survey” to assess what, exactly? I think you could elaborate and provide more concrete/actionable recommendations.

Reviewer #2: General Comments

The paper presents findings on a qualitative study to assess midwives understanding of respectful (and non-abusive) maternal care in a teaching hospital in Kumasi, Ghana. The aim is to use this information to design a facility-based intervention that addresses disrespectful care. Following 15 (or 16?) interviews, authors report 3 main themes and 11 sub-themes on midwives awareness of, motivation and recommendation for respectful maternal care. They conclude that midwives’ knowledge on RMC is high on all but one (i.e. evidence-based care) component.

The research design, study setting, data management and consent procedures are generally well described (pages 4-6). It seems that only one level of data analysis was conducted, making the thematic headings and reported data quite superficial. In the absence of in-depth critical analysis, the contribution of findings to the existing body of knowledge on RMC from the perspective of health workers, is quite limited. Although not reported, there are indications that a pre-existing lens or framework was applied in designing the interview tools and analyzing the data, leaving little room for new findings to emerge inductively. Furthermore, themes and subthemes overlap (eliminating distinction in meaning and interpretation) or are not linked. I found it quite interesting that while midwives claim to understand and support the rights of women during childbearing, the practice of slapping women’s thighs to encourage a smooth delivery was defended by some respondents.

Some of the conclusions and assertions presented need to be revised to match the modest study findings. In the move towards improving the quality of maternal care globally using a rights-based approach, respectful maternal care (RMC) has gained much needed attention. While there is certainly need for attention to be paid to RMC in research, policy and practice, I find that only the recommendation to improve in-service training for midwives is matched to the study findings. The paper does not report substantive information to justify a recommendation on the policy-front.

It is interesting to note that authors have not used this paper to build upon or reflect on previous work they also conducted on the same topic: Sarah D Rominski, Jody Lori, Emmanuel Nakua, Veronica Dzomeku, Cheryl A Moyer. When the baby remains there for a long time, it is going to die so you have to hit her small for the baby to come out": justification of disrespectful and abusive care during childbirth among midwifery students in Ghana. Health Policy and Planning, Volume 32, Issue 2, March 2017, Pages 215–224. The above was conducted in Ghana with midwifery students and reported similar findings, especially the wide gap between what is understood or known and what is practiced in RMC. Both studies therefore call to question the cliché recommendation that assumes training of health workers will sufficiently address RMC.

This makes me wonder the extent to which this paper has been situated in light of existing studies and other current literature.

Additional Comments

Major Issues

1. I would be hesitant to claim this paper is about knowledge of RMC, because standardized tools have not been used to measure their knowledge. Maybe better to stick to the words ‘awareness’, ‘understanding’ or ‘conceptualization’ as appropriate

2. I found the thematic analysis to be quite superficial. Themes and subthemes come across as overlapping or not linked, which eliminates distinction in meaning and interpretation. For example Table 1, page 8: 6 sub-themes are grouped under the theme “Awareness of RMC” (do you mean “Understanding of RMC”?). These are i) respectful care; ii) non-abusive care; iii) provision of child bearing women-centred care; iv) respecting child bearing women’s rights; v) evidence-based care and vi) illustrations of RMC

-Sub-themes i) and ii) as well as iii) and iv), are very similar and related. Could these not be grouped to produce two sub-themes? The sub-theme “illustrations of RMC” also seems disjointed from the main theme on awareness of RMC. Do the illustrations of RMC lead to or constitute an awareness of it?

-The term “provision of child bearing women-centred care” needs to be described or clarified. Is the goal women-centred care in general or women-centred care only for child bearing women? Are the distinctions relevant to this paper? This sub-theme is not connected to awareness of RMC.

-Are the 6 sub-themes expected to represent the dimensions of RMC according to this study or were these themes already prescribed during the study design?

3. It is reported that “All the midwives in the current study demonstrated some understanding of respectful and non-abusive maternal care.” (Page 8) Beyond reported information not confirmed by observation or a practical skills or knowledge tests, how was their understanding confirmed? ‘Awareness’ would be a more apt term.

4. Still on the issue of themes, the theme “Motivation for RMC” connotes an underlying assumption that respondents practice RMC, which was not ascertained by the study. This also make me curious to have a look at the interview guide to see how questions were posed.

5. The headings used to present findings as well as questions asked related to “the evidence-based component of maternity care” give the impression that an existing framework guided development of the interview guides and data analysis as opposed to the themes emerging inductively from the raw data. This should be clarified and explained. Furthermore, it could be useful to clarify what is meant by “the evidence-based component of maternity care”. According to which standards?

6. Following up on the previous comment, there seems to be some intention to distinguish evidence-based maternity care as a sub-component of respectful maternity care. See page 18, lines 360-363. Should respectful maternity care in its entirety not be evidence based? This buttresses the point that there is an unspoken lens/framework through which the study was approached. You probably want to add a paragraph in the introduction that clarifies the components or conceptualization of RMC.

7. Quotes should be included only to support and contribute additional insight to the narrative. There were many instances were quotes generally repeated what had already been described in the preceding paragraph (e.g. Page 10, quote from Midwife 007, Page 11, quote from Midwife 002) or two quotes were presented when one would have sufficed (Page 10). Also consider shortening quotes so they are more focused and to the point (e.g. page 4, quotes #1&2; page 10, last quote).

8. It is a bit strange that after reporting instances where midwives acknowledged physically hitting women during labor, authors report that midwives understood the non-violation of women’s rights (page 18, lines 360-362). This follows with stating that midwives have ‘substantial knowledge of RMC’ (page 18, line 364) although there was no data collected to support this conclusion.

9. Authors could have gone further in reflecting on the implications of findings. For example, the data shows a disconnection between the reported understanding/knowledge of the midwives and their actual practice (see lines 245-247 on page 13). The discussion is weak in situating study findings within the context of current literature. Given the goal of this sub-study to inform an intervention, it could have been interesting to read how these findings could be translated into the planned intervention.

Furthermore, on page 19, lines 370-377, factors that could promote RMC as identified by other studies have been mentioned, but these need to be situated within the paper itself. Were these factors also identified by respondents? To what extent could these (or lack of these) have played a role in shaping the findings.

The discussion ends quite abruptly (lines 385-391, page 19) following a missed opportunity to critique the recommendation midwives provided on pain and labor management i.e. beyond countering with a measure on training, which authors already acknowledge does not always translate into practice. Why do you think this was the lens through which they believe RMC would be addressed? Do midwives rationalize disrespectful care as being a consequence of women’s poor attitude and poor pain management? These are some of the questions I would have expected the discussion to analyse.

Minor Issues

1. The title reads “Exploring midwives’ understanding of respectful and non-abusive maternal care in Kumasi, Ghana: Qualitative Inquiry”. It seems clear that non-abusive care is implied in respectful maternal care. The title could therefore be revised to read: “Exploring midwives’ understanding of respectful maternal care in Kumasi, Ghana: Qualitative Inquiry”. If changed, please ensure it is streamlined and consistent throughout the paper.

2. If the paper focuses on RMC within the context of delivery/labour then the broad usage of the term “maternity care” should be revised and made specific. While the principles of RMC are similar through the continuum of care, the practical application of it could differ from antenatal to post-partum care.

3. Were participants’ consent secured in writing or verbally? Page 5, line 93- clarify what ‘officially agreed’ means.

4. Clarify the rationale for conducting only 15 interviews. Convenience? Data saturation?

Also note that while the paper mostly reports 15 interviews, on page 7, line 142-143, the total comes to 16 interviewees.

5. Consistency in describing what the paper intends to do is helpful. See page 4, line 74. Here is it reported as ‘experiences and views’. In other places, it is their ‘understanding’ (e.g. title) or “experiences and understanding” (Page 4, line 66)

6. Is it relevant to report the religion or parity of respondents in this paper? Was there an underlying hypothesis that these would influence their views on RMC?

7. Page 4, line 69-71 “Documenting the thoughts of midwives on RMC will help policymakers to address existing gaps in knowledge on RMC.” Can you clarify which persons constitute policy makers as used here? Do you mean at the parliamentary level as that may be quite far-reaching. The national midwifery council and training schools may be more appropriate policy actors in this scenario.

8. Page 4, line 73 “Qualitative research design with a phenomenological approach…” should be the other way around- “Phenomenological research design using a qualitative approach (or method)…..”

9. Page 5, line 92. Please clarify what specific expertise participants needed to have.

10. Page 5, line 101. Were the tools piloted in the same hospital where data was eventually collected?

11. Page 6, line 114. It would be helpful to elaborate a bit more on how thematic analysis principles were applied.

12. Page 12, line 233. So, what were their concerns?

13. Page 19, lines 386-387 “This recommendation is an example of how quick midwives are to outline measures directed at childbearing women to curtail D&AC.” Should there be a reference to this statement?

14. Page 20, lines 398-400. I would be hesitant to conclude that the paper provided “comprehensive information on midwives’ perspectives on RMC and the identification of skilled providers’ gap in knowledge on evidence-based maternity care”. One would need to review the interview guide used for its comprehensiveness and the data would have had to include a systematic knowledge assessment.

15. To aid replicability of your study, it could be helpful to include your interview guides as supplementary files.

16. Kindly check reference #15 (used on page 3, line 51). Since it is a study protocol, it cannot have yet shown the “effectiveness of RMC interventions……”

17. Complete citation information for reference #13, page 22.

Conclusion

In conclusion, while this article touches upon an interesting issue especially in light of the wider literature which it aims to contribute to (i.e. respectful maternity care and by extension, the quality of maternal care), it would benefit from being revised and restructured.

6. PLOS authors have the option to publish the peer review history of their article (what does this mean?). If published, this will include your full peer review and any attached files.

Reviewer #1: Yes: Kristy M. Hackett

Reviewer #2: Yes: Ibukun-Oluwa Abejirinde

---

## [Author Response · Author response to Decision Letter 0]

22 Nov 2019

Department of Nursing

College of Health Sciences

Kwame Nkrumah University of Science and Technology

Kumasi-Ghana

11th November 2019

Editorial Office

PLOS ONE

Dear Britt MacKinnon,

RE: INVITATION TO REVISE MANUSCRIPT PONE-D-19-19693

We express our appreciation to the editor and reviewers for the constructive critique of our manuscript.

In response to comments, we have provided a point by point response to each comment. Also, where substantive changes, deletions and insertion of key sentences have been made to the revised manuscripts, exact changes and where (pages/sections) effected, have been clearly stated as part of the responses to comments in the table.

In addition to this cover letter, we have provided a copy of the revised manuscript with tracked changes as required. Also, a clean revised manuscript without any tracked changes as required.

Thank you

Dzomeku Veronica Millicent

PRINCIPAL INVESTIGATOR & CORRESPONDING AUTHOR

Journal: PLOS ONE

Subject: Responses to Reviewers

REVIEWER 1

Abstract:

Comment: - While ensuring respectful maternal care is a critical goal in its own right, the rationale for this work may be further strengthened by stating the downstream implications of disrespect/abuse (for example, negative experience during childbirth may dissuade women from returning for facility-based postpartum services or from ANC/delivery for future pregnancies and births). It may be worth adding this rationale to the abstract if space allows.

- Suggest revising the methods to a more active voice (i.e. “we employed a phenomenological study design using in-depth interviews…”)

- What was the general analytical approach? If space allows, a brief sentence describing how transcripts were analyzed would be helpful.

- Line 32: What is meant by “gap in knowledge on evidence-based care”? This reads a little vague and the link to respectful maternal care is unclear. Could the authors be more specific here?

Response: We appreciate and acknowledged these useful comments and have revised our abstract to reflect the suggestions made by reviewer 1. The changes can be seen on page 1-2, line number 14-40.

Background:

Comment: - Line 39: suggest changing “recent times” to “recent years”

- Line 40: “undermining the purpose of encouraging childbearing…” is a little wordy and could be simplified.

- Lines 41- 42: Suggested edit: “rights to quality maternity care, including dignified care and freedom from discrimination”

- One thing I’m missing from the background/intro section is some indication or comment on why D&AC is so highly prevalent (is it because providers are overburdened, underpaid, etc. or is this completely lacking in medical training, etc. What does the evidence say?)

- Line 50: a nice review of relevant literature, but could the authors specify which outcomes are typically used to measure RMC? Line 60, for example: how are “women’s childbirth experiences measured?” Qualitatively? Or are there standard quantitative indicators? Including some detail on the metrics used will be helpful for readers not familiar with this area of work.

- Line 68: “evidence of incidences” is a bit awkward; could this be re-phrased?

- Line 70: suggest editing “further evidence of non-evidence based…” to “further qualitative insights into…”

Response: We acknowledge and find the comments by the reviewer 1 helpful, so we have updated our background section to reflect the grammatical inputs provided and have specifically provided a section on the outcomes that are typically used to measure RMC, how women’s childbirth experiences are measured, and the metrics used (please, see page 3, lines 54-59). We, however, did not provide a comment on why D&AC is so highly prevalent since it’s not directly related to our study’s objectives.

Design

Comments: - I think an overall “Methods” heading would be appropriate before other sections (before “Design”)

- Stating that a phenomenological approach was employed is a bit vague. Suggest elaborating with a few sentences to explain which phenomenological methods were applied, with appropriate citations.

- What were the specific research questions for this study? The overall aim (understanding midwives’ perspectives of RMC) reads a bit vague.

Response: We are grateful for the comments on our methodology and the kind recommendations offered because after having addressed them we have seen a facelift of the paper. We realized that all the suggestions made on the method, after addressing them, have satisfied many of the points in the 32-item checklist. Instead of justifying our choice of phenomenology, we rather chose to accept the suggestion made about the best method that suits our study. We purchase and used as the reference of the 2014 version of Green and Thorogood book which is cited accordingly in the manuscript [see page 4-5 line 85-90].

Setting

Comment: - What are the demographics of most maternity care patients in this facility? Are they mostly lower SES, are they diverse in terms of ethnicity, etc.? Some additional detail on the clients/patients served would be helpful.

Response: The authors have acknowledged this comment and have provided additional detail on clients/patients served on page 5, line numbers 96-98.

Population, sampling and sample size

Comment: - Purposive sampling: what kind of expertise was considered for interviewee selection? Don’t all midwives have relevant expertise?

- How were the study participants recruited? (approached by researchers directly?)

- Line 93: Sentence needs revising (“All participants who officially…”) the second clause about data saturation is out of place.

- How many providers were invited to participate? And of those, how many agreed to participate?

- The details about data/theme saturation should be moved to Data management and analysis (seems out of place in the sampling section)

Response: The authors have acknowledged these relevant questions and provided answers to them on page 5, line numbers 104-108.

Data collection

Comment: - How was the interview tool developed? Did researchers draw on any existing theory/frameworks to select relevant questions and probes?

- What settings were “suitable” for participants? Were most conducted within facilities?

Response: The authors acknowledged these critical questions and have expanded the data collection section of the manuscript by including answers to them (kindly check page 6, line number 110-130).

Data management and analysis

Comment- Were there any other quality assurance protocols in place? (e.g. de-briefing sessions post-interview, cross-checking transcripts?)

- Line 112: can delete “a qualitative data management software” (implied)

- How were the codes developed? Was this an iterative process – e.g. did they emerge from the data or did researchers apply codes based on topics covered in the interview guides?

- Was coding done as a team? How was consensus/agreement on coding reached?

- Line 118: Consider deleting the “trustworthiness/rigour” sub-heading and just include this paragraph under analysis. Alternatively, the sentences in this section could be re-distributed to the relevant sub-headings that precede this section.

- Line 121: What is meant by “member checking”; please define and elaborate and cite a method's source, if appropriate.

- “Independent analysis of the data” is quite vague – what exactly did authors do to cross-check/compare analytical rigour? (particularly when multiple analysis/coders were involved)

- Line 123: The statement about transferability is not convincing; providing sufficient detail is not sufficient to confirm transferability; suggest re-phrasing this.

- Some of the statements about adherence to the protocol and trustworthiness of data may be stated later (discussion section) in a strengths/limitations paragraph

- I think some reflection is needed on researcher positionality relative to interviewees (this is critical information for the assessment of qualitative studies).

Response: We consider these comments on the data management and analysis section of our manuscript very useful. These comments and questions guided the redrafting of this section, and it now reads better and has addressed all the comments and questions raised (kindly check page 6-7, line numbers 132-155).

Findings

Given that the comments on the findings and the discussions were many, and have addressed all of them to the best of our knowledge because of their relevance, we have chosen to provide brief responses to each of them by stating where in the manuscript those changes have been made.

Comments: - Line 142: what was the range in midwives’ years of practice? Were they all at the same level/rank?

Response: page 8, line 169

- Line 148: suggest to change to active voice

Response: page 9 line 175

- It would be helpful to see some examples of questions/probes used to elicit perspectives on RMC (could include the interview guide as a supplementary file, or at least give examples)

Response: We have added the interview guide as a supplementary file.

- Table 1 - It is not clear to me whether the themes and sub-themes truly “emerged” from the data organically, or if these topics were specifically probed (was the interview guide organized around these themes/sub-themes from the start?); I assume the latter, but this should be stated explicitly if so (and not labelled as “emergent”)

Response: page 6 line 110-114

- This reads more as a descriptive, thematic analysis (which is fine) – where does the phenomenological component come in? Phenomenology means different things to different researchers; as currently described, I’m not convinced it’s an appropriate label for the methodology used here.

Response: page 4-5 line 85-90

- Line 157-159: no need to duplicate information presented in table 1 and subsequent sub-headings (sub-themes are already clear in the table and deleting will save word count)

Response: we deleted table 1.

- Line 163: “these are illustrated…” can be deleted (implied that the quotations are meant to illustrate preceding statements)

Response: we deleted the phrase.

- I think the second quote is a better illustration of “respectful” care; are comfort measures synonymous with “respect”? Or just good clinical practice? Perhaps including definitions from the literature would be helpful under each of these types of care headings

Response: page 9 line 185-187

- A simpler coding system for the quotes could be used (since all participants are midwives, no need to state midwife every time)

Response: addressed.

- Line 178: “encapsulated providing… “ this sentence is wordy; could be simplified.

Response: we simplified the sentence: line 201

- Line 181: “desisting from” – change to “avoiding the use of...”

Response: line 204

- Were providers asked specifically to define/explain their understanding of these phrases (e.g. non-abusive care?)

Response: Yes, please. We have added the interview guide as a supplementary file for your perusal.

- Line 191: The first quote doesn’t add much (could delete or move to respectful care section), but the second is a nice illustration of women-centred care.

Response: we deleted the first quote. 

- The text in lines 204-206 could be condensed

Response: We condensed the text: line 223

- Line 211 (“the following are some of the views”); these lead-in sentences are a bit bulky and take up the precious word count. Is there a way to integrate relevant quotes into the text more smoothly?

Response: addressed and all lead in sentences have been deleted.

- Lines 228-229: again, it would be helpful to have a sense of which questions were asked

Response: we have added the interview guide as a supplementary file for your perusal.

- Line 227 down: the section on hitting women during active labour is important but I’m confused as to why this is included under “evidence-based care” heading and not, for example, respectful or non-abusive care. The link to evidence-based care seems like a stretch – perhaps the authors can better articulate how this component was conceptualized, or consider removing it completely.

Response: we revised this section and explain the rationale for treating it as a standalone theme: line 243-250 

- Line 269-280: this content seems somewhat out of place, and doesn’t flow from the preceding paragraphs. The authors may need to clarify how these illustrations specifically relate to RMC.

Response: we deleted this section as we have reviewed it and found it to be adding nothing relevant to our findings.

- Lines 287-207: my interpretation of this quote is that the midwife is motivated by the perceived connection between how the mother is treated and neonatal outcomes (more so than the wellbeing of the woman); perhaps revisit this interpretation or select a different quote?

Response: We have corrected and used the suggested interpretation: line 269-274 

- Page 16 – the selected quotes are appropriate and interesting, but three may not be necessary; could condense.

Response: one quote deleted.

- Line 337-338: this text is not needed

Response: we deleted the text.

- Line 339 down (labour and pain management education): what’s interesting here is that there’s an implicit blaming of women for the abusive/disrespectful treatment (they don’t know how to do what is expected of them during labour, so midwives become tired and frustrated); if only they knew better… Shouldn’t respectful care be the standard, regardless of women’s knowledge of labour expectations? This might be something to consider elaborating on in the discussion/interpretation (you speak to it briefly in lines 385-387); it’s one of the more interesting and novel findings presented.

Response: line 316-321

- It occurs to me, when I get to the end of the Findings section, that there was no clear framework used to guide analysis/interpretation. Are there standard conceptual frameworks and definitions of RMC that could be usefully applied in this paper? I think framing/structuring the results around current theoretical models/conceptual framework would strengthen the paper.

Response: The interview questions (with the probes) were asked in accordance with the framework developed from the first author’s preliminary work (cited in the revised manuscript on line 112). We considered this comment useful and have restructured the result section along with the major themes in the interview guide.

Discussion (singular, remove “s” from the heading)

Comments: - First sentence can be deleted (actually, the whole first paragraph could be omitted)

Response: we deleted the whole paragraph

- Lines 360-363: Based on the findings presented, I’m not convinced that they completely understood each component of RMC; perhaps rephrase this statement to be less definitive.

Response: we rephrased the statement: line 337-340

- There is only one example of “non-evidence based care” given in the results section (hitting), so drawing conclusions about gaps in midwives’ knowledge is not well substantiated.

Response: addressed: line 337-340

- Overall, the discussion is quite sparse and would be strengthened with a more critical interpretation of the findings, and emphasis on what this study specifically adds to the growing body of work in this area.

Response: the entire discussion section has benefited from the revision: line 337-370

- Additional recommendations for policy and practice would also strengthen this section

Response: line 385-388

- To what extent are these ideals (i.e. aspects of RMC identified by midwives) actually practised? Did the transcripts give any indication of whether respectful care is actually implemented? It is possible that midwives have good knowledge/awareness of what it means to provide respectful care, but this doesn’t mean best practices are actually implemented. This would be important to comment on in the discussion section.

Response: line 374-377

- Line 393: delete “obvious”

Response: deleted

- Line 394: Unclear how “exploratory nature” of the study limits its generalizability (this has more to do with the small sample size and is typical of qualitative studies of this nature)

Response: rephrased: line 372-374

- Perhaps one limitation is that you are unable to confirm whether the reported best practices are actually implemented; future studies could build on this work by assessing the quality of care delivered in this setting (via clinical observation or by interviewing women receiving care)

Response: line 374-377

- Line 395-397: re “bias” this is not so much a limitation (you did not intend to assess objective measures of RMC delivered, but rather the midwives’ lived experience/personal opinions and viewpoints). I think this section could be re-considered more critically, and then revised.

Response: the statement on bias has been removed.

- Line 409: “nation-wide cross-sectional quantitative survey” to assess what, exactly? I think you could elaborate and provide more concrete/actionable recommendations

Response: we addressed this on line numbers 385-388.

REVIEWER 2

The reviewer 2 provided very useful and critical comments and questions and broadly categorized them under two main subheadings: major issues and minor issues. Considering the fact that the comments and questions were many, and having addressed every issue raised in the revised manuscript, we decided to rather indicate where the corrections were made in the manuscript. Thanks a lot.

Major Issues

1. I would be hesitant to claim this paper is about knowledge of RMC, because standardized tools have not been used to measure their knowledge. Maybe better to stick to the words ‘awareness’, ‘understanding’ or ‘conceptualization’ as appropriate

Response: throughout the manuscript

2. I found the thematic analysis to be quite superficial. Themes and subthemes come across as overlapping or not linked, which eliminates distinction in meaning and interpretation. For example Table 1, page 8: 6 sub-themes are grouped under the theme “Awareness of RMC” (do you mean “Understanding of RMC”?). These are i) respectful care; ii) non-abusive care; iii) provision of childbearing women-centred care; iv) respecting childbearing women’s rights; v) evidence-based care and vi) illustrations of RMC

-Sub-themes i) and ii) as well as iii) and iv), are very similar and related. Could these not be grouped to produce two sub-themes? The sub-theme “illustrations of RMC” also seems disjointed from the main theme on awareness of RMC. Do the illustrations of RMC lead to or constitute an awareness of it?

Response: line 175-177

-The term “provision of childbearing women-centred care” needs to be described or clarified. Is the goal women-centred care in general or women-centred care only for childbearing women? Are the distinctions relevant to this paper? This sub-theme is not connected to the awareness of RMC.

Response: line 212

-Are the 6 sub-themes expected to represent the dimensions of RMC according to this study or were these themes already prescribed during the study design?

Response: line 144-145

3. It is reported that “All the midwives in the current study demonstrated some understanding of respectful and non-abusive maternal care.” (Page 8) Beyond reported information not confirmed by observation or practical skills or knowledge tests, how was their understanding confirmed? ‘Awareness’ would be a more apt term.

Response: line 337

4. Still on the issue of themes, the theme “Motivation for RMC” connotes an underlying assumption that respondents practice RMC, which was not ascertained by the study. This also makes me curious to have a look at the interview guide to see how questions were posed.

Response: We added the interview guide as a supplementary file for your perusal.

5. The headings used to present findings as well as questions asked related to “the evidence-based component of maternity care” give the impression that an existing framework guided development of the interview guides and data analysis as opposed to the themes emerging inductively from the raw data. This should be clarified and explained. Furthermore, it could be useful to clarify what is meant by “the evidence-based component of maternity care”. According to which standards?

Response: line 243-250

6. Following up on the previous comment, there seems to be some intention to distinguish evidence-based maternity care as a sub-component of respectful maternity care. See page 18, lines 360-363. Should respectful maternity care in its entirety not be evidence-based? This buttresses the point that there is an unspoken lens/framework through which the study was approached. You probably want to add a paragraph in the introduction that clarifies the components or conceptualization of RMC.

Response: line 243-250

7. Quotes should be included only to support and contribute additional insight to the narrative. There were many instances were quotes generally repeated what had already been described in the preceding paragraph (e.g. Page 10, quote from Midwife 007, Page 11, quote from Midwife 002) or two quotes were presented when one would have sufficed (Page 10). Also, consider shortening quotes so they are more focused and to the point (e.g. page 4, quotes #1&2; page 10, last quote).

Response: the first quote deleted.

8. It is a bit strange that after reporting instances where midwives acknowledged physically hitting women during labour, authors report that midwives understood the non-violation of women’s rights (page 18, lines 360-362). This follows by stating that midwives have ‘substantial knowledge of RMC’ (page 18, line 364) although there was no data collected to support this conclusion.

Response: addressed: line 337-340

9. Authors could have gone further in reflecting on the implications of findings. For example, the data shows a disconnection between the reported understanding/knowledge of the midwives and their actual practice (see lines 245-247 on page 13). The discussion is weak in situating study findings within the context of current literature. Given the goal of this sub-study to inform intervention, it could have been interesting to read how these findings could be translated into the planned intervention.

Response: line 343-348.

Furthermore, on page 19, lines 370-377, factors that could promote RMC as identified by other studies have been mentioned, but these need to be situated within the paper itself. Were these factors also identified by respondents? To what extent could these (or lack of these) have played a role in shaping the findings.

Response: we deleted this paragraph.

The discussion ends quite abruptly (lines 385-391, page 19) following a missed opportunity to critique the recommendation midwives provided on pain and labour-management i.e. beyond countering with a measure on training, which authors already acknowledge does not always translate into practice. Why do you think this was the lens through which they believe RMC would be addressed? Do midwives rationalize disrespectful care as being a consequence of women’s poor attitude and poor pain management? These are some of the questions I would have expected the discussion to analyse.

Response: line 361-370 

Minor Issues

1. The title reads “Exploring midwives’ understanding of respectful and non-abusive maternal care in Kumasi, Ghana: Qualitative Inquiry”. It seems clear that non-abusive care is implied in respectful maternal care. The title could, therefore, be revised to read: “Exploring midwives’ understanding of respectful maternal care in Kumasi, Ghana: Qualitative Inquiry”. If changed, please ensure it is streamlined and consistent throughout the paper.

Response: line 1

2. If the paper focuses on RMC within the context of delivery/labour then the broad usage of the term “maternity care” should be revised and made specific. While the principles of RMC are similar through the continuum of care, the practical application of it could differ from antenatal to postpartum care.

Response: fixed: line 76

3. Were participants’ consent secured in writing or verbally? Page 5, line 93- clarify what ‘officially agreed’ means.

Response: addressed: line 107

4. Clarify the rationale for conducting only 15 interviews. Convenience? Data saturation?

Also note that while the paper mostly reports 15 interviews, on page 7, line 142-143, the total comes to 16 interviewees.

Response: fixed

5. Consistency in describing what the paper intends to do is helpful. See page 4, line 74. Here is it reported as ‘experiences and views’. In other places, it is their ‘understanding’ (e.g. title) or “experiences and understanding” (Page 4, line 66)

Response: fixed.

6. Is it relevant to report the religion or parity of respondents in this paper? Was there an underlying hypothesis that these would influence their views on RMC?

Response: We added them simply as demographic characteristics of respondents.

7. Page 4, line 69-71 “Documenting the thoughts of midwives on RMC will help policymakers to address existing gaps in knowledge on RMC.” Can you clarify which persons constitute policymakers as used here? Do you mean at the parliamentary level as that may be quite far-reaching. The national midwifery council and training schools may be more appropriate policy actors in this scenario.

Response: line 79-82

8. Page 4, line 73 “Qualitative research design with a phenomenological approach…” should be the other way around- “Phenomenological research design using a qualitative approach (or method)…..”

Response: line 85-90

9. Page 5, line 92. Please clarify what specific expertise participants needed to have.

Response: line 104-108

10. Page 5, line 101. Were the tools piloted in the same hospital where data was eventually collected?

Response: fixed: line 114-117

11. Page 6, line 114. It would be helpful to elaborate a bit more on how thematic analysis principles were applied.

Response: fixed: line 140-141

12. Page 12, line 233. So, what were their concerns?

Response: fixed: line 244-250

13. Page 19, lines 386-387 “This recommendation is an example of how quick midwives are to outline measures directed at childbearing women to curtail D&AC.” Should there be a reference to this statement?

Response: fixed, and no reference needed as it’s our interpretation of the findings: line 362-364 

14. Page 20, lines 398-400. I would be hesitant to conclude that the paper provided “comprehensive information on midwives’ perspectives on RMC and the identification of skilled providers’ gap in knowledge on evidence-based maternity care”. One would need to review the interview guide used for its comprehensiveness and the data would have had to include a systematic knowledge assessment.

Response: we revised this section and have added the interview guide for your perusal. Line 381-388.

15. To aid the replicability of your study, it could be helpful to include your interview guides as supplementary files.

Response: We added the interview guide as a supplementary file for your perusal.

16. Kindly check reference #15 (used on page 3, line 51). Since it is a study protocol, it cannot have yet shown the “effectiveness of RMC interventions……”

Response: removed

17. Complete citation information for reference #13, page 22.

Response: fixed: line 434-436

Journal Requirements:

Response: Fixed

a) Did participants provide their written or verbal informed consent to participate in this study?

Response: line 107

3. Please include a copy of the interview guide used in the study, in both the original language and English, as Supporting Information, or include a citation if it has been published previously."

Response: we have added a copy of the interview guide for your perusal.

4. We noted in your submission details that a portion of your manuscript may have been presented or published elsewhere: "The methods section was taken from a manuscript submitted to the BMC Pregnancy and Childbirth Journal. This does not constitute dual publication because the objectives of the two papers are different."

Response: We addressed this concern in the cover letter.

We are grateful for all the comments and questions provided by the reviewers and the academic editor. Thanks a lot.

---

## [Decision Letter · Decision Letter 1]

11 May 2020

PONE-D-19-19693R1

Exploring midwives’ understanding of respectful maternal care in Kumasi, Ghana: Qualitative Inquiry

PLOS ONE

Dear Dzomeko Veronica Millicent ,

Thank you for submitting your manuscript to PLOS ONE. After careful consideration, we feel that it has merit but does not fully meet PLOS ONE’s publication criteria as it currently stands. Therefore, we invite you to submit a revised version of the manuscript that addresses the points raised during the review process.

We would appreciate receiving your revised manuscript by 12th June. To enhance the reproducibility of your results, we recommend that if applicable you deposit your laboratory protocols in protocols.io, where a protocol can be assigned its own identifier (DOI) such that it can be cited independently in the future. For instructions see: http://journals.plos.org/plosone/s/submission-guidelines#loc-laboratory-protocols

We look forward to receiving your revised manuscript.

Kind regards,

Sharon Mary Brownie

Academic Editor

PLOS ONE

Additional Editor Comments (if provided):

Congratulations on a well received manuscript. Please attend to the recommendations from the reviewers and return your revised manuscript by the due date.

Reviewers' comments:

Reviewer's Responses to Questions

**Comments to the Author**

Reviewer #1: (No Response)

Reviewer #3: All comments have been addressed

2. Is the manuscript technically sound, and do the data support the conclusions?

Reviewer #1: Yes

Reviewer #3: (No Response)

3. Has the statistical analysis been performed appropriately and rigorously? 

Reviewer #1: N/A

Reviewer #3: (No Response)

4. Have the authors made all data underlying the findings in their manuscript fully available?

Reviewer #1: No

Reviewer #3: Yes

5. Is the manuscript presented in an intelligible fashion and written in standard English?

Reviewer #1: No

Reviewer #3: Yes

6. Review Comments to the Author

Reviewer #1: Reviewer 1 comments (round 2)

Thank you again for the opportunity to review this important work. The authors should be commended for their work on the revisions. Overall, the paper is much improved, and has the potential for publication pending some additional editing and revisions. I note some additional suggestions below:

Abstract:

- line 16: can delete "childbearing women's" to start sentence with “Experience of…”

- Line 20: text would flow better if the sentence “Midwives’ knowledge of…” came before the preceding sentence.

- Line 38: First sentence of the conclusion section can be deleted (redundant – this is stated in the sentence that follows)

- Lines 39-41 (“Despite their awareness…”) these sentences can be deleted; new results (including specific examples) should be presented in the results section, not conclusion. The conclusion can really start at “In view of these findings”, since the first few sentences are already clearly presented in the results section.

- Line 43: delete “that there should be”

Methods:

- Line 95: delete “explorative” – you say explore later in the senetence.

- Lines 149-151: This description is similar to the one for the other author (lines 39-41 above) so reads as repetitive. Can the info be combined into one sentence to avoid repetition in the text?

- Lines 161-162: “Each transcript was coded…” This sentence is redundant (this is stated above). In general, the manuscript could be more concise throughout by cutting repetitive sentences.

- Lines 172-174: Transferability is the degree to which the qualitative findings might also apply to other contexts/populations, but this text seems to describe replicability of the study itself. Suggest to delete this sentence or re-work it.

- Looking at the interview guide, it seems as though the authors sought to answer a specific question from the outset (i.e. IS there a disconnect between midwives’ awareness of RMC, and their actual practice). If this is the case, it should be stated in the methods section and some indication of how questions were designed to address this question should be included.

Findings

- Line 279 - I like the interpretation re: disconnect between awareness and actual practice, but to me this is more of an interpretation/finding of the study, as opposed to a standalone theme. The authors might consider re-phrasing this section. Also, this section seems somewhat out of place; it may be better to cover these findings last, in order to transition into discussion of the apparent disconnect.

- Line 298: unclear as to what is meant by “evidence-based” here

- Lines 402-404: “This recommendation suggests…” While I agree with this interpretation, I think it may be better saved for the discussion section rather than results.

Discussion

- remove “s” on Discussions header

- line 24: typo – delete “and” after respectful

- Line 425: “non-evidence based” seems like the wrong word here (and throughout) – could you just say “disrespectful and/or abusive” practices?

- First paragraph: it’s clear that the study findings are in line with previous research in Ghana. One of the criteria for publication is whether the paper adds new evidence – what is different about this study, and what novel findings add to existing literature on the subject? Some consideration of this would strengthen the paper considerably.

- There are some interesting, practical recommendations for ways to improve RMC, which are grounded in the results – this is great to see.

- Do the authors have any thoughts on other constraints to RMC that may not have been probed in the interviews? For example, is it possible that midwives are overworked/under-resourced and therefore rushing during childbirth? While disrespectful/abusive treatment is never acceptable under any circumstance, I do wonder whether there are health system challenges that contribute to healthcare workers’ inability to practice RMC.

- Overall, the discussion is a bit sparse, and could be strengthened with additional commentary on how this work adds to existing literature, and what opportunities it highlights for future research.

Overall comments:

- The paper generally reads well, although some additional copyediting is needed to improve the flow of the text.

- I am unsure about data availability – authors state that all data is presented in the manuscript, but only select quotations are included. If PLOS requires full transcripts to be made available, then authors may need to revisit this statement.

Reviewer #3: Thank you for the opportunity to review this manuscript under title Exploring midwives’ understanding of respectful maternal care in Kumasi, Ghana: Qualitative Inquiry

I think the 1st time the reviewers covered the manuscript with the helpful and scientific comments, and Authors response to comments evidently and correcting the comments scientifically.

Only I have two points:

Abstract

It is more scientific if not use the abbreviations in the abstract. Like RMC. Write down the full meaning. Page 2, in line 32, 33,35.

Findings

Check the numbers of midwives in the findings part.

its fifteen midwives ???? only check.

Reviewer #1: Yes: Kristy M. Hackett

Reviewer #3: No

---

## [Author Response · Author response to Decision Letter 1]

30 May 2020

Response to reviewers

We appreciate the comments and suggestions of the reviewers. We have revised the manuscript in line with the comments and the suggestions.

Reviewer #1: Reviewer 1 comments (round 2)

Abstract:

- line 16: can delete "childbearing women's" to start sentence with “Experience of…”

- Line 20: text would flow better if the sentence “Midwives’ knowledge of…” came before the preceding sentence.

- Line 38: First sentence of the conclusion section can be deleted (redundant – this is stated in the sentence that follows)

- Lines 39-41 (“Despite their awareness…”) these sentences can be deleted; new results (including specific examples) should be presented in the results section, not conclusion. The conclusion can really start at “In view of these findings”, since the first few sentences are already clearly presented in the results section.

- Line 43: delete “that there should be”

Response: We are grateful for these comments on the abstract. We have made the changes, and they can be found on lines 18-19, 30-31.

Methods:

- Line 95: delete “explorative” – you say explore later in the senetence.

Response: We have deleted it, and the new sentence with the correction is on line 81.

- Lines 149-151: This description is similar to the one for the other author (lines 39-41 above) so reads as repetitive. Can the info be combined into one sentence to avoid repetition in the text?

Response: We have address the issue on line 128-129 as follows:

 ”has same educational and research experience as the second author in addition to midwifery education.”

- Lines 161-162: “Each transcript was coded…” This sentence is redundant (this is stated above). In general, the manuscript could be more concise throughout by cutting repetitive sentences.

Response: We have gone through the manuscript and cut out repetitive sentences from the revise manuscript. Regarding the specific revision requested, we have rewritten the sentence to read as follows on line 137: “The codes generated during the analyses were synthesized into predetermined subthemes.”

- Lines 172-174: Transferability is the degree to which the qualitative findings might also apply to other contexts/populations, but this text seems to describe replicability of the study itself. Suggest to delete this sentence or re-work it.

Response: We have addressed this comment on line 139 and 144 by replacing ”transferability” with ”replicability”

- Looking at the interview guide, it seems as though the authors sought to answer a specific question from the outset (i.e. IS there a disconnect between midwives’ awareness of RMC, and their actual practice). If this is the case, it should be stated in the methods section and some indication of how questions were designed to address this question should be included.

Response: We are greatful for this comment. Our main research objective is to explore midwive’s awareness of RMC at the study setting. The issue of disconnect between knowledge and practice was a secondary objective, and the reason for pursuing this has been mentioned and explained on line as follows 307-310: ” The views of midwives were sought on situations where force should be applied during intrapartum care. This question was informed by findings from previous work by the first, third, and fifth authors involving midwifery students who justified the application of force or hitting childbearing women during labour [34].”

Findings

- Line 279 - I like the interpretation re: disconnect between awareness and actual practice, but to me this is more of an interpretation/finding of the study, as opposed to a standalone theme. The authors might consider re-phrasing this section. Also, this section seems somewhat out of place; it may be better to cover these findings last, in order to transition into discussion of the apparent disconnect.

Response: We are grateful for this comment. We have rearrange the result section and brought the findings on the disconnect between awareness and actual practice to the last section of the result section. These changes can be found on line 306-326.

- Line 298: unclear as to what is meant by “evidence-based” here

Response: Thank you for your comment. We have replaced ”evidence-based” with ”disrespect and abusive practices” in line with your comment on sections of our discussion.

-Lines 402-404:“This recommendation suggests…” While I agree with this interpretation, I think it may be better saved for the discussion section rather than results.

Response: We have address this comment and move this interpretation to the discussion section on line 355-357

Discussion

- remove “s” on Discussions header

Response: we have address this comment. Thank you.

- line 24: typo – delete “and” after respectful

Response: we have address this comment. Thank you.

- Line 425: “non-evidence based” seems like the wrong word here (and throughout) – could you just say “disrespectful and/or abusive” practices?

Response: we have address this comment. Thank you.

- First paragraph: it’s clear that the study findings are in line with previous research in Ghana. One of the criteria for publication is whether the paper adds new evidence – what is different about this study, and what novel findings add to existing literature on the subject? Some consideration of this would strengthen the paper considerably.

- There are some interesting, practical recommendations for ways to improve RMC, which are grounded in the results – this is great to see.

Response: Thank you for the compliment.

- Do the authors have any thoughts on other constraints to RMC that may not have been probed in the interviews? For example, is it possible that midwives are overworked/under-resourced and therefore rushing during childbirth? While disrespectful/abusive treatment is never acceptable under any circumstance, I do wonder whether there are health system challenges that contribute to healthcare workers’ inability to practice RMC.- Overall, the discussion is a bit sparse, and could be strengthened with additional commentary on how this work adds to existing literature, and what opportunities it highlights for future research.

Response: We have address this comment and the one on paragraph by adding a full paragrah addressing all the suggestions made. These changes are on line 364-378 as follows:

 ” Research on the implications of the disconnect between knowledge and practice in healthcare are amply documented, with complex systems such as motivational, institutional, and sociocultural factors cited as barriers to applying knowledge. Given that disrespectful and abusive care within the context of labour and childbirth has recently started receiving critical scholarly appraisal, not many studies have document the disconnect that exist between midwives’ knowledge of respectful maternity care practices and what they practice in the labour ward. Our study highlight this disconnect. Therefore, this finding call for research opportunities that probe this issue further and explore provider and institutional level factors that may be driving the disconnect in knowing what to do to promote optimal labour and childbirth experience to childbearing women and the practice of disrespectful care. While disrespectful and abusive practices in maternity care is never acceptable under any circumstance, the call for further research to explore drivers can as a matter of urgency examine whether health system challenges contribute to healthcare workers’ inability to practice RMC in the study setting. With findings from studies guided by these recommended research questions, policymakers will be equipped to design programs and policies that holistically address disrespectful care in intrapartum care.”

Overall comments:

- The paper generally reads well, although some additional copyediting is needed to improve the flow of the text.

Response: We are grateful for your comment. We have done further proofing of the manuscript in line with your comment.

- I am unsure about data availability – authors state that all data is presented in the manuscript, but only select quotations are included. If PLOS requires full transcripts to be made available, then authors may need to revisit this statement.

Response: We have updated our data availability to address this comment.

Reviewer #3: Thank you for the opportunity to review this manuscript under title Exploring midwives’ understanding of respectful maternal care in Kumasi, Ghana: Qualitative Inquiry

I think the 1st time the reviewers covered the manuscript with the helpful and scientific comments, and Authors response to comments evidently and correcting the comments scientifically.

Only I have two points:

Abstract

It is more scientific if not use the abbreviations in the abstract. Like RMC. Write down the full meaning. Page 2, in line 32, 33,35.

Response: we have address this comment. We have written RMC in full as Respectful maternity care. Thank you.

Findings

Check the numbers of midwives in the findings part.

its fifteen midwives ???? only check.

Response: Thank you for the comment. The study involved 15 midwives. The quotes in the results sections are taken from transcripts that are clear and presentable under the themes. Therefore, quote from all respondents are not present in the manuscript.

Thank you.

---

## [Editor Report · Decision Letter 2]

10 Jun 2020

Exploring midwives’ understanding of respectful maternal care in Kumasi, Ghana: Qualitative Inquiry

PONE-D-19-19693R2

Dear Dr.Dzomeku Veronica Millicent,

We’re pleased to inform you that your manuscript has been judged scientifically suitable for publication and will be formally accepted for publication once it meets all outstanding technical requirements.

Kind regards,

Sharon Mary Brownie

Academic Editor

PLOS ONE

Additional Editor Comments (optional):

Reviewer recommendations have been addressed.

---

## [Editor Report · Acceptance letter]

29 Jun 2020

PONE-D-19-19693R2 

Exploring midwives’ understanding of respectful maternal care in Kumasi, Ghana: Qualitative Inquiry 

Dear Dr. D-zomeku:

I'm pleased to inform you that your manuscript has been deemed suitable for publication in PLOS ONE. Congratulations! Your manuscript is now with our production department. 

Kind regards, 

on behalf of

Professor Sharon Mary Brownie 

Academic Editor

PLOS ONE